# Risk of recurrent venous thromboembolism in patients with HIV infection: A nationwide cohort study

**Casper Rokx**[1☯]*, **Jaime F. Borjas Howard**[2☯], **Colette Smit**[3], **Ferdinand W. Wit**[4], **Elise D. Pieterman**[1], **Peter Reiss**[4], **Suzanne C. Cannegieter**[5], **Willem M. Lijfering**[5], **Karina Meijer**[2], **Wouter Bierman**[6], **Vladimir Tichelaar**[2], **Bart J. A. Rijnders**[1], on behalf of the ATHENA observational HIV cohort[¶]

1 Erasmus MC, University Medical Centre Rotterdam, Department of Internal Medicine, Section of Infectious Diseases, Rotterdam, the Netherlands, 2 University of Groningen, University Medical Centre Groningen, Department of Haematology, Groningen, the Netherlands, 3 HIV Monitoring Foundation, Amsterdam, the Netherlands, 4 Department of Global Health and Division of Infectious Diseases, Amsterdam University Medical Centres, University of Amsterdam, Amsterdam, the Netherlands, 5 Leiden University Medical Centre, Department of Clinical Epidemiology, Leiden, the Netherlands, 6 University of Groningen, University Medical Centre Groningen, Department of Internal Medicine, Infectious Diseases Service, Groningen, the Netherlands

☯ These authors contributed equally to this work.
¶ Membership of the ATHENA observational HIV cohort is provided in S1 Supplementary Methods.
* c.rokx@erasmusmc.nl

**Data Availability Statement:** Data sharing where it concerns data of people living with HIV (PWH) has been restricted by the Medisch Ethische Toetsingscommissie (Institutional Review Board)

## Abstract

### Background

Multiple studies have described a higher incidence of venous thromboembolism (VTE) in people living with an HIV infection (PWH). However, data on the risk of recurrent VTE in this population are lacking, although this question is more important for clinical practice. This study aims to estimate the risk of recurrent VTE in PWH compared to controls and to identify risk factors for recurrence within this population.

### Methods and findings

PWH with a first VTE were derived from the AIDS Therapy Evaluation in the Netherlands (ATHENA) cohort (2003–2015), a nationwide ongoing cohort following up PWH in care in the Netherlands. Uninfected controls were derived from the Multiple Environmental and Genetic Assessment of risk factors for venous thrombosis (MEGA) follow-up study (1999–2003), a cohort of patients with a first VTE who initially participated in a case-control study in the Netherlands who were followed up for recurrent VTE. Selection was limited to persons with an index VTE suffering from deep vein thrombosis in the lower limbs and/or pulmonary embolism (PE). Participants were followed from withdrawal of anticoagulation to VTE recurrence, loss to follow-up, death, or end of study. We estimated incidence rates, cumulative incidence (accounting for competing risk of death) and hazard ratios (HRs) using Cox proportional hazards regression, adjusting for age, sex, and whether the index event was

of the Academic Medical Center of the University of Amsterdam because the data from PWH underlying this study contain very sensitive and potentially identifying information. Requests for data sharing can nonetheless be made on a case-by-case basis following submission of a concept sheet as per instructions on the website of the ATHENA cohort (https://www.hiv-monitoring.nl/en/research-using-our-data/submit-research-proposal) and from MEGA, by requesting data from the Department of Clinical Epidemiology from the Leiden University Medical Center (Datamanager: Ingeborg de Jonge, Data management office, Department of Clinical Epidemiology C7-P, Leiden University Medical Center, P.O. Box 9600, 2300 RC Leiden, The Netherlands, email: i.de_jonge@lumc.nl).

**Funding:** The authors received no specific funding for this work. The MEGA study is supported by the Dutch Heart Foundation (grants NHS98.113, NHS2010B167, NHS208B086, and NHS2011T012), the Dutch Cancer Foundation (RUL 99/1992), and the Netherlands Organization for Scientific Research (grant 912-03-033|2003). The ATHENA cohort is supported by a grant from the Dutch Ministry of Health, Welfare and Sport through the Centre for Infectious Disease Control of the National Institute for Public Health and the Environment. Funding in both studies was not linked to any individual author. The funders had no role in study design, data collection and analysis, decision to publish, or preparation of the manuscript.

**Competing interests:** I have read the journal's policy and the authors of this manuscript have the following competing interests: CR reports grants from Gilead, Merck, and personal fees from Gilead, ViiV, Janssen-Cilag. JFBH and EDP report no conflicts of interest. YIGVT was employed by the Dutch National Health Institute during analysis and reporting on this study. CS reports grants from Netherlands Ministry of Health, Welfare and Sport, National Institute for Public Health and the Environment, Centre for Infectious Disease Control, during the conduct of the study. FW reports personal fees from Gilead Sciences, personal fees from ViiV Healthcare, outside the submitted work. PR reports independent scientific grant support to his institution from Gilead Sciences, Janssen Pharmaceuticals Inc., Merck & Co., Bristol-Myers Squibb, and ViiV Healthcare; fees to his institution for his participation on scientific advisory boards for Gilead Sciences ViiV Healthcare, Merck & Co., Teva pharmaceutical industries, and on a data safety monitoring committee for Janssen Pharmaceuticals Inc. BR reports grants from

provoked or unprovoked. When analyzing risk factors among PWH, the main focus of analysis was the role of immune markers (cluster of differentiation 4 [CD4]+ T-cell count).

There were 153 PWH (82% men, median 48 years) and 4,005 uninfected controls (45% men, median 49 years) with a first VTE (71% unprovoked in PWH, 34% unprovoked in controls) available for analysis. With 40 VTE recurrences during 774 person-years of follow-up (PYFU) in PWH and 635 VTE recurrences during 20,215 PYFU in controls, the incidence rates were 5.2 and 3.1 per 100 PYFU (HR: 1.70, 95% CI 1.23–2.36, $p = 0.003$). VTE consistently recurred more frequently per 100 PYFU in PWH in all predefined subgroups of men (5.6 versus 4.8), women (3.6 versus 1.9), and unprovoked (6.0 versus 5.2) or provoked (3.1 versus 2.1) first VTE. After adjustment, the VTE recurrence risk was higher in PWH compared to controls in the first year after anticoagulant discontinuation (HR: 1.67, 95% CI 1.04–2.70, $p = 0.03$) with higher cumulative incidences in PWH at 1 year (12.5% versus 5.6%) and 5 years (23.4% versus 15.3%) of follow-up. VTE recurred less frequently in PWH who were more immunodeficient at the first VTE, marked by a better CD4+ T-cell recovery on antiretroviral therapy and during anticoagulant therapy for the first VTE (adjusted HR: 0.81 per 100 cells/mm$^3$ increase, 95% CI 0.67–0.97, $p = 0.02$). Sensitivity analyses addressing potential sources of bias confirmed our principal analyses. The main study limitations are that VTEs were adjudicated differently in the cohorts and that diagnostic practices changed during the 20-year study period.

## Conclusions

Overall, the risk of recurrent VTE was elevated in PWH compared to controls. Among PWH, recurrence risk appeared to decrease with greater CD4+ T-cell recovery after a first VTE. This is relevant when deciding to (dis)continue anticoagulant therapy in PWH with otherwise unprovoked first VTE.

## Author summary

### Why was this study done?

- The HIV pandemic affects approximately 40 million people and causes significant morbidity, including a markedly increased risk of a venous thromboembolism (VTE).
- The recurrence risk of VTE in people living with HIV (PWH) is unknown, although this risk drives the anticoagulant therapy duration after a first VTE.
- Our study determined the recurrent VTE risk in PWH compared to uninfected controls.

### What did the researchers do and find?

- We performed an observational cohort study using data from the national ATHENA PWH cohort (2003–2015) in the Netherlands and the Dutch Multiple Environmental

Gilead, grants from MSD, nonfinancial support from MSD, nonfinancial support from Gilead, nonfinancial support from BMS, nonfinancial support from Janssen-Cilag, nonfinancial support from ViiV, nonfinancial support from Abbvie, personal fees from Gilead, personal fees from ViiV, personal fees from Great-Lakes pharmaceuticals, outside the submitted work; and financial compensation payed to institution for advisory board participation organized by Gilead, ViiV, BMS, Janssen-Cilag, and MSD. KM received research support from Bayer, Sanquin, and Pfizer; speaker fees from Bayer, Sanquin, Boehringer Ingelheim, BMS, and Aspen; consulting fees from Uniqure (outside the submitted work, all fees go to the institution). WBW reports reimbursement payed to institution for investigator-initiated study from Janssen-Cilag, financial compensation payed to institution for multicenter study by GSK and catering of a symposium by Janssen-Cilag, all outside the submitted work. SCC is a member of the Editorial Board of *PLOS Medicine*. WML reports no conflicts of interest.

**Abbreviations:** ATHENA, AIDS Therapy Evaluation in the Netherlands; cART, combination antiretroviral therapy; CD4, cluster of differentiation 4; DOAC, direct oral anticoagulant; DVT, deep venous thrombosis; HR, hazard ratio; ICD-10, International Classification of Diseases, 10th revision; IVDU, intravenous drug use; MEGA, Multiple Environmental and Genetic Assessment of risk factors for venous thrombosis; PE, pulmonary embolism; PWH, people with HIV; PYFU, person-years of follow-up; STROBE, Strengthening the Reporting of Observational Studies in Epidemiology; VTE, venous thromboembolism.

and Genetic Assessment of risk factors for venous thrombosis (MEGA) cohort (1999–2009) of HIV-uninfected controls with a first VTE.

- The recurrent VTE incidence rate per 100 person-years of follow-up (PYFU) was higher in PWH (5.2) compared to controls (3.1) yielding a 1.70 hazard ratio (HR; 95% CI 1.23–2.36). Incidence rates were consistently higher for PWH in subgroups stratified by sex or the cause of the first VTE.

- PWH with lower cluster of differentiation 4 (CD4)+ T-cell counts at their first VTE had fewer recurrent events, which was driven by PWH experiencing a better CD4+ T-cell recovery on HIV treatment prior to anticoagulant discontinuation.

### What do these findings mean?

- The risk of recurrent VTE is apparently increased in PWH but is ameliorated with better immune reconstitution.

- HIV-associated immunodeficiency reflects a reversible risk factor for VTE specific to PWH and is of relevance for decisions on anticoagulant therapy duration.

## Introduction

An infection with HIV results in an increased risk of a first venous thromboembolism (VTE) [1]. This is likely related to the observed procoagulant state in people with HIV (PWH) [2–4] and is in line with the increased first VTE rates associated with other infections [5–8]. Indeed, multiple cohorts found a 2- to 10-fold increased risk of a first VTE in PWH compared to the general population. Additionally, among PWH, studies have shown that the risk of VTE was higher if PWH had lower plasma cluster of differentiation 4 (CD4)+ T-cell counts, had evidence of viremia, or had clinically active opportunistic infections [9–11]. With the increasing prevalence of HIV worldwide, now approaching 40 million patients globally, awareness of VTE risk in PWH is important [12].

The clearly established increased risk of a first VTE is in stark contrast with the lack of reliable data regarding the risk of a subsequent recurrent VTE in PWH, although this information is crucial to determine the optimal duration of anticoagulant therapy after a first VTE. Generally speaking, when any patient with a first VTE has completed 3 months of anticoagulant therapy, the treating physicians should weigh the risk for VTE recurrence against major bleeding complications associated with prolonged use [13]. The recurrence risk is estimated by the presence of provoking and potentially reversible risk factors during the first VTE [14]. Patients with a provoked first VTE due to a persistent risk factor (e.g., metastatic cancer) are considered to have the highest recurrence risk. Patients without an identifiable risk factor have an unprovoked VTE, with an intermediate risk for recurrence. Patients with a provoked first VTE due to a transient risk factor are at lowest risk for recurrence. Transient risk factors that have clearly been associated with a lower risk of recurrence are surgery, plaster cast immobilisation and/or use of oestrogen-containing contraceptives. In patients with a presumed low risk of bleeding, current guidelines advocate withdrawal of anticoagulants only if the index VTE was associated with a transient provoking risk factor [13].

Considering this framework in thinking about the recurrence risk spectrum, it is important to understand how HIV infection influences this recurrence risk. One line of reasoning may be that current combination antiretroviral therapy (cART) regimens only suppress viral replication without curing the disease. Hence, a VTE associated with HIV infection can be seen as associated with a persistent risk factor, placing such patients on the high end of the recurrence risk spectrum. A more nuanced view would be that the recurrence risk may depend on the changing underlying disease status induced by cART: PWH with ongoing immune deficiency, viremia, and/or opportunistic infections are likely to be in a procoagulant state and therefore may have a high risk of recurrence. In turn, PWH who had a first VTE associated with these sequelae but have recovered on cART may (partially) reverse the initial procoagulant state, translating to a low risk of recurrence. The reasoning that transient inflammation may exert the same prognostic implications as other established transient VTE risk factors is supported by a recent study showing that people suffering a VTE associated with bacterial infections have a lower risk of recurrence. In this study, the recurrence risk was equivalent to the risk associated with established transient provoking factors [15]. If reversal of HIV-specific factors by use of cART indeed shows a similar risk of recurrence as these factors, then limited duration of anticoagulation should be considered in such patients.

In summary, the risk of recurrent VTE in PWH is essentially unknown. We hypothesize that overall, the risk of recurrent VTE will be higher in PWH than in HIV-uninfected controls. Also, we hypothesize that the risk is influenced by HIV-specific factors—most importantly immune status and viral load—and therefore recurrence risk may be higher if disease is not controlled and low when HIV-specific factors present during a first VTE have been reversed. This study therefore aims to investigate the risk of recurrent VTE in PWH by comparing this risk to a group of uninfected controls and to explore whether HIV-specific factors influence the risk of recurrence.

## Methods

### Study design

We used 2 historical Dutch cohorts: the national AIDS Therapy Evaluation in the Netherlands (ATHENA) observational HIV cohort [16] and the Multiple Environmental and Genetic Assessment of risk factors for venous thrombosis (MEGA) follow-up study [17]. ATHENA was approved by the institutional review board of all participating centres. People entering HIV care receive written material about participation in the ATHENA cohort and are informed by their treating physician of the purpose of data collection, after which they can consent verbally or opt-out—which is possible at any later time. The MEGA study was approved by the institutional review board of the Leiden University Medical Centre, and written informed consent was obtained by from all participants. No specific consent was obtained for the current study as the data were analyzed anonymously. Study protocols were reviewed by scientific boards from both study centres. The study protocol regarding the current analysis is available as Supporting Information (S1 Protocol).

### Participants

In both cohorts, included patients were ≥18 years with a nonfatal first VTE, being a deep venous thrombosis (DVT) of the legs starting in the popliteal vein or more proximal and/or a pulmonary embolism (PE). To be able to determine the recurrence risk and answer the question most important to clinical practice (what is the risk of recurrence if anticoagulation is withdrawn?), included participants had to have discontinued anticoagulant therapy after a first

VTE. To mirror the current treatment guideline paradigm, anticoagulation should have been administered for at least 3 months.

In ATHENA, all Dutch HIV treatment centres participate, and over 98% of all PWH provide consent for their data to be used for research purposes [16]. The present study was conducted in the 12 largest centres, making up 70% of all PWH in care in the Netherlands. PWH with a first VTE were identified as described previously [11]. In summary, we employed a case-finding strategy to detect cases through registered use of anticoagulants by Anatomical Therapeutic Chemical codes (see S1 Supplementary Methods). This strategy was first evaluated in a pilot and had 100% sensitivity for detecting lower limb DVT and PE. First VTE were detected from the period January 2003, reflecting the start of anticoagulant registration in ATHENA, until April 2015. All VTE were adjudicated using radiological or clinical reports mentioning specific anatomical locations, or if at least 3 months of anticoagulant treatment was given for diagnosed DVT or PE without specific reports on anatomical locations. Trained data collectors validated all suspected cases on-site.

Details of the MEGA follow-up study have been described previously [17–19]. In short, 4,956 consecutive patients aged ≤70 years with a first VTE were recruited between March 1999 and August 2004 in 6 anticoagulant clinics in a case-control study. These clinics were responsible for the anticoagulant care of all VTE patients within a well-defined area in the Netherlands. Patients completed an extensive questionnaire on putative VTE risk factors. The diagnosis of a first VTE was adjudicated either through radiological confirmation or through a recorded diagnosis in the participating anticoagulation clinics. The participants from this study were then subsequently followed up by contacting them between June 2008 and July 2009 (further ascertainment of outcome outlined subsequently). PWH in this study were identified through registered cART use and subsequently excluded.

## Exposures

In both cohorts, first VTE were categorized as provoked or unprovoked. In ATHENA, provoked VTE was associated with either a cancer diagnosis or anticancer treatment within 180 days (excluding basal/squamous skin cell carcinomas), surgery, pregnancy/puerperium, oestrogen contraceptive exposure, leg fracture with plaster cast, hospitalization, and/or immobilization for >3 days within 90 days prior to the VTE. The period 9 months before to 3 months after the delivery date defined the pregnancy/puerperium. In PWH, these exposures of interest and relevant variables were collected using standardized case report forms. To assess HIV-specific factors for VTE, we extracted from the ATHENA database all registered data on plasma HIV RNA and CD4+ T-cell counts, cART use, and opportunistic infections.

Two provoking factors were registered differently in MEGA: both any VTE that occurred in a 5-year period after a cancer diagnosis and all VTE occurring in a 90-day period following hospitalisation, regardless of duration, were considered provoked. We performed sensitivity analyses to determine the effect on risk estimates of these 2 differing definitions (details following). All other decision rules to define provoked first VTE, baseline variables, and potential confounders related to VTE were similar to ATHENA.

Of note, none of the PWH patients were treated for their VTE with direct oral anticoagulants (DOACs). None of the MEGA individuals either were treated with DOACs as DOACs were not available during the observation period.

## Outcomes

In both cohorts, recurrent VTE were classified as either certain or possible, according to decision rules employed previously in the MEGA follow-up study (see S1 Supplementary

Methods). Certain recurrent VTE were scored in case of evidently new and clinically well-documented events. The primary endpoint in all analyses was the incidence of any certain recurrent VTE.

In ATHENA, information about VTE recurrences was collected by reviewing the electronic medical records of PWH with a first VTE on-site by the trained data collectors. For PWH who had been previously in care at any other HIV treatment centre during any VTE, the ATHENA network ensured the possibility to query data managers in these centres for information on radiology reports and clinical notes. Recurrent VTE were registered until September 2018.

In MEGA, 225 of 4,956 patients did not consent to follow-up for VTE recurrence [17]. To identify recurrences, the vital status of participants was ascertained in the central Dutch population register, and causes of death (International Classification of Diseases, 10th revision [ICD-10]) were obtained from the national register of death certificates at the Central Bureau of Statistics. Questionnaires on recurrent VTE were mailed to all survivors, complemented by telephone interviews. Additionally, the necessary information regarding recurrence was acquired from the anticoagulation clinics and hospitals. All participants were assessed between 2007 and July 2009 by these 3 approaches to ensure correct adjudication of recurrent VTE.

## Statistical analysis

In all analyses, the index date for start of follow-up was the date of anticoagulant discontinuation. Censoring occurred at loss to follow-up, non–VTE-related death, last visit, or study end date, whatever occurred first. Participants who had a possible recurrent VTE were censored at the date of this recurrence. Additionally, a post hoc decision was made to exclude follow-up time beyond 6 years. This was decided because of the sparse numbers in the PWH cohort still at risk at this time point and because the cumulative incidence of recurrent VTE at around the 5-year mark is used to inform the decision to continue or withhold anticoagulant treatment [20].

Crude incidence rates were calculated by dividing the number of VTE by 100 person-years of follow-up (PYFU) overall and by subgroups known to be important for prognosis (sex, provoked versus unprovoked index VTE). Kaplan-Meier analyses of VTE recurrence accounted for competing risk of death by assigning a separate censoring code to persons with a non–VTE-related death. Plots were stratified by sex and provoked versus unprovoked first VTE. Cox proportional hazards regression was used to estimate adjusted hazard ratios (HRs) for recurrent VTE. Models were adjusted for age, sex, and whether the index event was provoked or unprovoked. The proportional hazard assumption was assessed by visually inspecting plots and inspecting Schoenfeld residuals.

In the analysis of HIV-specific risk factors, we specifically aimed to determine the effects of reversible risk factors between first VTE and anticoagulation withdrawal, hypothesizing that index events associated with risk factors that were reversed would be associated with a lower risk of recurrence. Reversibility of risk factors was therefore operationalized as change of a given marker between index VTE and anticoagulation withdrawal. For CD4+ T-cell count, this was done by extracting the CD4+ T-cell count closest to the date of anticoagulant withdrawal and subtracting this by an extracted CD4+ T-cell count closest to the VTE index date. An identical approach was planned for calculating change in viral load. However, after inspection of descriptive data, this analysis was abandoned because there was insufficient variability in the change in viral load due to most participants having a suppressed plasma viral load at the index VTE or close thereafter.

Given limitations of sample size within PWH, a limited amount of degrees of freedom were available for a Cox model to include change in CD4+ T-cell count as a category as well as

adjustment for essential confounders. Beyond adjustment for age, sex, and unprovoked VTE, HR estimates for the association of CD4+ T-cell count change with recurrent VTE were also adjusted for the presence of an intercurrent infection at the time of the first VTE; such infections are known to further lower CD4+ T-cell counts and therefore may cause a rapid CD4 + T-cell recovery when treated. Therefore, the change in CD4+ T-cell count was modelled as a continuous variable in the Cox regression model. However, to ease interpretability, incidence rates and Kaplan Meier plots were categorized per quartile and presented separately.

Finally, to explore what proportion of recurrence burden for PWH might be attributable to these risk factors, at the time of recurrence we describe the immune status and viral load of PWH who suffered a recurrence.

## Sensitivity analyses

Sensitivity analyses were performed to evaluate possible sources of bias. These analyses were all done post hoc. The main sources were potential misclassification of confounders and outcome misclassification due to different validation methods between the cohorts.

The impact of differing definitions of hospitalization as provoked/unprovoked VTE was explored in 2 ways: (1) reclassifying all VTEs associated with hospitalization in ATHENA as provoked VTE and subsequently refitting regression models and vice versa and (2) reclassifying provoked VTEs in MEGA (which had only hospitalization as a provoking factor) as unprovoked and refitting regression models.

The impact of differing definitions of cancer-associated VTE between the 2 cohorts was analysed by simulating a worst-case misclassification scenario. Considering that the definition of cancer-associated VTE used in MEGA was broad (any cancer present 5 years compared to 180 days prior to index VTE in ATHENA) a proportion of the index events in MEGA defined as provoked by cancer would in fact be unprovoked—leading to incorrect adjustment of the effect of HIV because the effect of unprovoked versus provoked is potentially mis-specified. In the worst case, misclassification would have occurred exactly in participants that suffered a recurrence. This scenario was thus simulated by reclassifying participants in the control group who had cancer as the only provoking factor and suffered a recurrence as having had an unprovoked index event.

Biases in outcome ascertainment were explored by re-estimating HRs considering a composite outcome including possible VTE as well as certain VTE. Also, due to the difference in follow-up years, trends in diagnostic practice may have led to relative overadjudication of recurrent VTEs in the PWH cohort due to diagnosis, most tangibly due to diagnosis of subsegmental PE [21]. In a sensitivity analysis, we censored observations relating to recurrent VTEs diagnosed as isolated subsegmental PE from the PWH cohort. We did not have this information available from MEGA, but data suggest that the influence of this was negligible [22]—so we sufficed by only addressing this in the PWH cohort. Furthermore, we assessed for less tangible period effects by restricting the analysis to only the follow-up years that overlapped (2003–2009).

In the analysis of HIV-specific risk factors, the influence of the assumptions made when constructing changes in CD4+ T-cell counts were explored. Importantly, some of the data may have been prone to immortal time bias: for some participants, CD4+ T-cell counts registered after withdrawal of anticoagulation were used in the analysis—if this CD4+ T-cell count was the measurement closest to the withdrawal date. Measurement of such a CD4+ T-cell count may have been influenced by occurrence of a recurrent VTE. In a sensitivity analysis, extraction of laboratory data was restricted to only measurements done prior to the date of anticoagulation withdrawal.

Furthermore, in patients with high CD4+ T-cell counts (>500) and on cART at the index date, fluctuations of CD4+ T-cell counts are not reflective of immune recovery (nor of a deterioration of their immune system in case the fluctuation is in the opposite direction) and are considered of no significance in clinical practice. Therefore, models were rerun considering only participants with a CD4+ T-cell count below 500 cells/mm$^3$ at the time of index VTE. By limiting this particular analysis to the subgroup of PWH who had CD4+ T-cell counts below 500 at the time of first VTE, increase in signal-to-noise ratio is expected.

Several analyses were suggested by peer reviewers to support the consistency of our results. These are discussed in Supporting Information (see S1 Supplementary Analyses).

Statistical analysis was performed using the survival package included in R, version 3.5.0. Reporting of this study has been done by the authors in accordance with Strengthening the Reporting of Observational Studies in Epidemiology (STROBE) guidelines for cohort studies. A completed checklist is available in Supporting Information (see S1 STROBE Checklist).

## Results

In a total of 14,389 PWH (99,762 PYFU), 203 PWH with a first leg DVT and/or PE were identified as described previously [11]. Of these, 25 did not withdraw anticoagulants after 6 months' treatment, 1 reached end of study before 6 months' treatment, 18 died before being treated 6 months, 3 were lost to follow-up before 6 months' treatment had passed, and in 2 other participants, it was unclear whether there had been at least 3 months of anticoagulant therapy use. This left 153 PWH available for the analysis. Of 4,731 patients in MEGA, 715 did not withdraw anticoagulants, and 11 were HIV infected, leaving 4,005 patients for the analysis.

Baseline characteristics are shown in Table 1. The groups appeared balanced regarding age, first VTE location, and anticoagulation duration. Both had a median 6 years of follow-up. Unlike controls, the majority of PWH had an unprovoked first VTE (71% versus 34%). The proportion of male PWH was 82%, resembling demographics of the national epidemic. At their first VTE, occurring at median 5 years after HIV diagnosis, the majority of PWH (68%) were on cART, 57% had plasma HIV RNA < 50 copies/mL, and the median CD4+ T-cell count was 410/mm$^3$. PWH died more frequently during follow-up (11% versus 1%).

### Rates and cumulative incidence estimates

Overall, 40 certain recurrent VTE in PWH and 635 certain recurrent VTE in HIV-uninfected controls were identified, yielding crude incidence rates of 5.2/100 PYFU and 3.1/100 PYFU, respectively (Table 2). Consistent with prior literature, rates were higher in men compared to women and in participants with an unprovoked index VTE as opposed to those with a provoked index event. Within these strata, the recurrence rate was higher in PWH compared to controls.

Recurrent VTE Kaplan-Meier estimates were higher in PWH compared to controls in all analyses (Fig 1, S1 Fig and S2 Fig). Overall, the difference in cumulative incidence was higher at each at time point, although the difference in absolute cumulative incidence stabilises after 2 years: at 1 year following anticoagulant withdrawal, the cumulative incidences were 12.5% (95% CI 8.2%–18.9%) versus 5.6% (95% CI 4.9%–6.3%), respectively. At 2 years, the cumulative incidences were 17.2% (95% CI 12.0%–24.2%) versus 8.9% (95% CI 7.8%–9.7%), and at 5 years, the cumulative incidences were 23.4% (95% CI 17.3%–31.3%) versus 15.3% (95% CI 14.1%–16.5%).

**Table 1. Baseline characteristics of participants at the time of their first VTE.**

| Characteristic | PWH (*n* = 153) | Controls (*n* = 4,005) |
|---|---|---|
| **Male sex, *n* (%)** | 126 (82%) | 1,813 (45%) |
| **Age at first VTE, median years (IQR)** | 48 (42–57) | 49 (38–58) |
| **Location of first VTE** | | |
| PE (with/without DVT elsewhere), *n* (%) | 75 (49%) | 1,637 (41%) |
| Proximal leg DVT, *n* (%) | 78 (51%) | 2,368 (59%) |
| **Unprovoked first VTE, *n* (%)** | 108 (71%) | 1,361 (34%) |
| **Provoked first VTE, *n* (%)** | 45 (29%) | 2,644 (66%) |
| Index event associated with surgery, *n* (%) | 4 (3%) | 666 (17%) |
| Index event associated with a cancer diagnosis, *n* (%) | 12 (8%) | 238 (6%) |
| Index event associated with pregnancy/puerperium, *n* (%) | 9 (6%) | 155 (4%) |
| Index event associated with use of contraceptives, *n* (%) | 1 (0%) | 1,200 (30%) |
| Index event associated with being bedridden, *n* (%) | 13 (8%) | 1,040 (26%) |
| Index event associated with lower leg immobilisation, *n* (%) | 12 (8%) | 600 (15%) |
| Index event associated with hospitalization, *n* (%) | 39 (25%) | 679 (17%) |
| Missing, *n* (%) | 0 (0%) | 104 (3%) |
| **Duration of anticoagulation, months (IQR)** | 6 (6–7) | 6 (4–7) |
| **Follow-up, median years (IQR)** | 5.7 (3.2–8.7) | 6.1 (4.5–8.7) |
| **Censoring due to loss to follow-up/end of study, *n* (%)** | 100 (65%) | 3,319 (83%) |
| **Censored due to non–VTE-related death, *n* (%)** | 17 (11%) | 46 (1%) |
| **HIV-specific characteristics** | **PWH *n* = 153** | |
| **Transmission mode** | | |
| MSM, *n* (%) | 76 (50%) | |
| Heterosexual, *n* (%) | 47 (31%) | |
| IVDU, *n* (%) | 8 (5%) | |
| Other, *n* (%) | 2 (1%) | |
| Unknown, *n* (%) | 20 (13%) | |
| **Time since HIV diagnosis, median years (IQR)** | 5.0 (1.0–10.7) | |
| **On cART, *n* (%)** | 104 (68%) | |
| **Concomitant clinical infection, *n* (%)[1]** | 31 (20%) | |
| **Prior CDC B or C event, *n* (%)** | 21 (14%) | |
| **CD4+ T-cell count, median cells/mm$^3$ (IQR)** | 410 (232–585) | |
| <200, *n* (%) | 51 (33%) | |
| 200–349, *n* (%) | 32 (21%) | |
| 350–499, *n* (%) | 28 (18%) | |
| ≥500, *n* (%) | 42 (28%) | |
| **CD4/CD8 ratio, median (IQR)** | 0.50 (0.26–0.80) | |
| **HIV RNA, median copies/mL (IQR)** | <50 (<50–5,028) | |
| <50, *n* (%) | 87 (57%) | |
| 50–999, *n* (%) | 20 (13%) | |
| 1,000–99,999, *n* (%) | 21 (14%) | |
| ≥100,000, *n* (%) | 21 (14%) | |
| Missing, *n* (%) | 4 (2%) | |

[1]Clinical infection refers to the presence of a symptomatic (opportunistic) infection other than HIV.

**Abbreviations:** cART, combination antiretroviral therapy; CD4, cluster of differentiation 4; CDC, Centers for Disease Control and Prevention; DVT, deep vein thrombosis; IQR, interquartile range; IVDU, intravenous drug use; MSM, men having sex with men; PE, pulmonary embolism; PWH, people with HIV; VTE, venous thromboembolism

**Table 2. Incidence rates in PWH versus controls.**

| Variable | PWH | | | Controls | | |
|---|---|---|---|---|---|---|
| | Events | PYFU | Rate/100 (95% CI) | Events | PYFU | Rate/100 (95% CI) |
| Overall | 40 | 774 | 5.2 (3.8–7.0) | 635 | 20,215 | 3.1 (2.9–3.4) |
| Male sex | 34 | 607 | 5.6 (4.0–7.8) | 410 | 8,523 | 4.8 (4.4–5.3) |
| Female sex | 6 | 167 | 3.6 (1.5–7.5) | 225 | 11,692 | 1.9 (1.7–2.2) |
| Unprovoked first VTE | 33 | 548 | 6.0 (4.2–8.4) | 338 | 6,522 | 5.2 (4.7–5.8) |
| Provoked first VTE | 7 | 226 | 3.1 (1.4–6.1) | 283 | 13,268 | 2.1 (1.9–2.4) |

Abbreviations: PWH, people with HIV; PYFU, person-years of follow-up; VTE, venous thromboembolism

## Cox regression results

Table 3 shows the results of Cox regression analysis comparing PWH to controls. Considering follow-up until 6 years, the unadjusted Cox regression model yielded an HR of 1.81 (95% CI 1.29–2.53). This was attenuated to 1.22 (0.87–1.73) after adjustment for age, sex, and provoked versus unprovoked index event. However, as suggested by the Kaplan-Meier plots, the proportional hazards assumption was violated in the model considering the total follow-up period (Schoenfeld residual *p*-value for HIV covariate: 0.05). We therefore decided to split the Cox models at the 1-year time point. After adjusting for age, sex, and provoked versus unprovoked first VTE (Table 2), PWH remained at higher risk for recurrent VTE compared to controls during the first year following anticoagulant withdrawal (HR 1.67, 95% CI 1.04–2.70), but the relative hazard became comparable thereafter (HR 0.94, 95% CI 0.58–1.54).

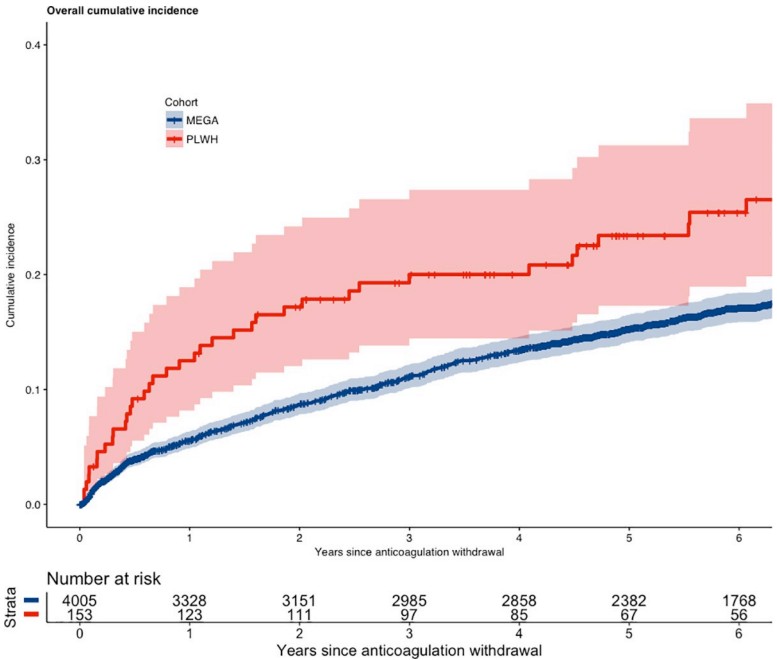

**Fig 1. Overall cumulative incidence of recurrent VTE in PWH and controls (MEGA).** MEGA, Multiple Environmental and Genetic Assessment of risk factors for venous thrombosis; PLWH, people living with HIV; VTE, venous thromboembolism.

**Table 3. Results of Cox regression models comparing PWH to controls.**

| Time period | Crude HR (95% CI) | Adjusted HR (95% CI) | p | Unprovoked aHR (95% CI) | p |
|---|---|---|---|---|---|
| Overall | 1.70 (1.23–2.36) | 1.16 (0.84–1.62) | 0.37 | 1.18 (0.81–1.70) | 0.39 |
| Follow-up up to 6 years | 1.81 (1.29–2.53) | 1.22 (0.87–1.73) | 0.25 | 1.21 (0.82–1.77) | 0.33 |
| Follow-up up to 1 year | 2.39 (1.50–3.83) | 1.67 (1.04–2.70) | 0.03 | 1.81 (1.09–3.03) | 0.02 |
| Follow-up from 1 to 6 years | 1.41 (0.87–2.30) | 0.94 (0.58–1.54) | 0.81 | 0.82 (0.46–1.48) | 0.51 |

Cox regression models stratified by time period, considering all events and unprovoked index VTEs only.

Adjusted models: adjusted for age, sex, and provoked versus unprovoked index VTE.

**Abbreviations:** aHR, adjusted hazard ratio; HR, hazard ratio; PWH, people with HIV; VTE, venous thromboembolism

## HIV-specific factors

A better CD4+ T-cell recovery in the time between diagnosing the first VTE and anticoagulant withdrawal was associated with a lower VTE recurrence (Table 4), which persisted over time (Fig 2). In adjusted models, the recurrence risk in PWH was lower in PWH that were more immunodeficient at the time of their first VTE. This lower risk was predicted by a better subsequent CD4+ T-cell recovery (Table 3). Analyzing the effect of viremia on VTE recurrence was not meaningful given the low number of PWH without suppressed plasma HIV RNA in the time period between diagnosis of index VTE and cessation of anticoagulation.

Among PWH who suffered recurrent VTE, over 65% had a CD4+ T-cell count over 500, and 83% had an undetectable viral load at the time of their recurrence.

## Sensitivity analyses

Table 5 shows results of sensitivity analyses. To evaluate the robustness of the findings, we explored the effects of possible misclassification. The main source of potential bias was the difference in definitions regarding hospitalisation and cancer. Reclassification of first VTE

**Table 4. Crude recurrent VTE rates and HRs according to absolute CD4 T-cell count at index VTE and CD4 T-cell count recovery.**

| Quartiles of ΔCD4 T-cell count | Events | PYFU | Rate/100 PYFU (95% CI) |
|---|---|---|---|
| First quartile ($< -66$ CD4+ T cells/mm$^3$) | 13 | 150 | 8.7 (4.8–14.5) |
| Second quartile ($-66$ to $+5$ CD4+ T cells/mm$^3$) | 14 | 142 | 9.9 (5.6–16.2) |
| Third quartile ($+5$ to $+110$ CD4+ T cells/mm$^3$) | 6 | 200 | 3.0 (1.2–6.2) |
| Fourth quartile ($> +110$ CD4+ T cells/mm$^3$) | 7 | 203 | 3.5 (1.5–6.9) |
| | Unadjusted HR (95% CI) | Adjusted HR (95% CI) | p |
| CD4+ T-cell count at first VTE (per 100 cells/mm$^3$ lower) | 0.89 (0.82–0.97) | 0.93 (0.84–1.02) | 0.12 |
| ΔCD4+ T-cell count (per 100 cells/mm$^3$ increase)[1] | 0.81 (0.69–0.95) | 0.81 (0.68–0.97) | 0.01 |

[1]The ΔCD4+ T-cell count refers to the change in CD4+ T cells between the first VTE and anticoagulant discontinuation. Adjusted HR: adjusted for age, sex, provoked versus unprovoked first VTE and clinical infection at first VTE. Clinical infection refers to the presence of a symptomatic (opportunistic) infection other than HIV (see Table 1).

**Abbreviations:** CD4, cluster of differentiation 4; HR, hazard ratio; PWH, people with HIV; PYFU, person-years of follow-up; VTE, venous thromboembolism

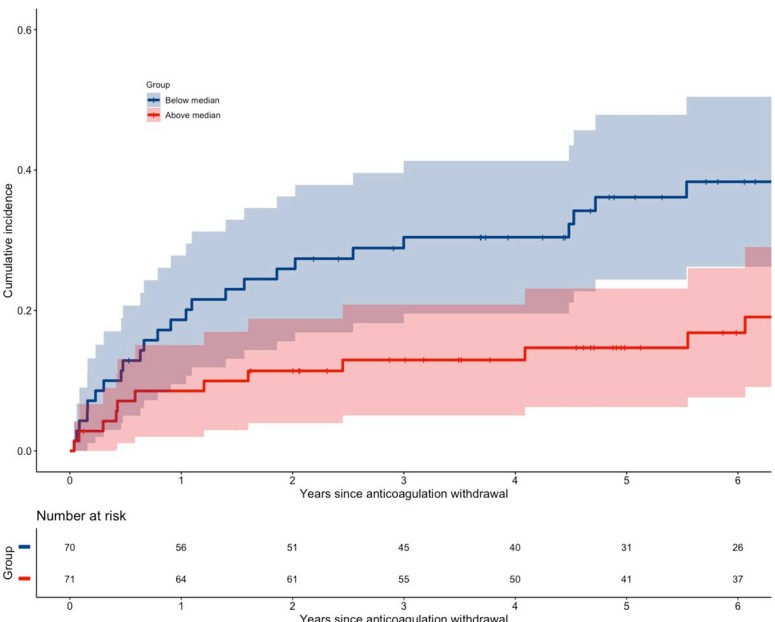

**Fig 2. Cumulative incidence of recurrent VTE in PWH split at median ΔCD4+ T-cell count.** CD4, cluster of differentiation 4; PWH, people with HIV; VTE, venous thromboembolism.

associated with hospitalisation or cancer did not have major influences on the effect estimates between PWH and controls. When considering both certain and possible recurrences as the composite outcome, the HRs for <1 year of follow-up were slightly attenuated overall, with the confidence interval in the adjusted model including 1 (HR: 1.40, 95% CI 0.88–2.22). With regard to overadjudication in the PWH cohort due to an isolated subsegmental PE as a recurrence: there was only one recurrence involving subsegmental PE, occurring at 2-year follow-up. This did not materially change results. The sensitivity analysis restricting analysis of cohorts within overlapping years of follow-up is shown in Supporting Information (see S1 Supplementary Analyses); HR estimates were consistent with the main analysis.

**Table 5. Sensitivity analyses of Cox regression examining recurrent VTE in PWH versus controls.**

| | HR (95% CI) Original Model* | HR (95% CI) Cancer Sensitivity | HR (95% CI) Hospitalisation Sensitivity 1* | HR (95% CI) Hospitalisation Sensitivity 2* | HR (95% CI) + Possible Recurrence | HR (95% CI) Subsegmental PE Excluded |
|---|---|---|---|---|---|---|
| **Overall** | 1.16 (0.84–1.62) | 1.15 (0.82–1.60) | 1.22 (0.88–1.70) | 1.19 (0.85–1.66) | 1.07 (0.78–1.46) | 1.13 (0.81–1.58) |
| **Follow-up < 6 years** | 1.22 (0.87–1.73) | 1.21 (0.86–1.70) | 1.28 (0.91–1.80) | 1.25 (0.89–1.76) | 1.11 (0.81–1.54) | 1.18 (0.84–1.68) |
| **Follow-up < 1 year** | 1.67 (1.04–2.70) | 1.66 (1.03–2.68) | 1.75 (1.09–2.82) | 1.70 (1.05–2.74) | 1.40 (0.88–2.22) | 1.67 (1.04–2.70) |
| **Follow-up 1–6 years** | 0.94 (0.58–1.55) | 0.93 (0.57–1.52) | 0.99 (0.61–1.62) | 0.97 (0.60–1.59) | 0.93 (0.59–1.46) | 0.89 (0.53–1.47) |

*All models adjusted for age, sex, provoked versus unprovoked VTE. Hospitalisation Sensitivity Model 1 recategorizes all first VTEs associated with hospitalisation as provoked first VTEs. Hospitalisation Sensitivity Model 2 recategorizes all first VTEs associated with hospitalisation as the only provoking factor as unprovoked first VTE.

**Abbreviations:** HR, hazard ratio; PE, pulmonary embolism; PWH, people with HIV; VTE, venous thromboembolism

**Table 6. Sensitivity analysis of Cox regression of HIV-specific factors on the risk of recurrent VTE.**

| | HR (95% CI) Hospitalisation Sensitivity | HR (95% CI) Certain and Possible Recurrent VTE | HR (95% CI) CD4+ Specific Sensitivity 1 | HR (95% CI) CD4+ Specific Sensitivity 2 |
|---|---|---|---|---|
| ΔCD4+ T-cell count (per 100 cells/mm³ increase) | 0.81 (0.67–0.98) | 0.84 (0.71–1.01) | 0.73 (0.59–0.88) | 0.68 (0.49–0.95) |

All models adjusted for age, sex, unprovoked versus provoked index VTE, and infection at first VTE. Clinical infection refers to the presence of a symptomatic (opportunistic) infection other than HIV (see Table 1). Sensitivity 1: restricted to CD4+ T-cell measurements before withdrawal of anticoagulation. Sensitivity 2: restricted to PWH with a CD4+ T-cell count < 500 cells/mm³ at the time of index VTE.

**Abbreviations:** CD4, cluster of differentiation 4; HR, hazard ratio; PWH, people with HIV; VTE, venous thromboembolism

With regard to the sensitivity analyses exploring assumptions around change in CD4+ T-cell count (Table 6), restricting to data only available prior to withdrawal of anticoagulation or to only participants with CD4+ counts below 500 at index VTE did not change the direction of the effect estimates. Indeed, the effect of CD4+ count recovery became slightly stronger.

## Discussion

In this study, we found that PWH were at higher risk of recurrent VTE compared to HIV-uninfected controls, particularly in the first year after withdrawing anticoagulants. Second, recurrence risk was lower in PWH with better immune recovery over time, which is more likely to occur in PWH with low CD4+ T-cell counts compared to PWH with already well reconstituted CD4+ T-cell counts at their first VTE.

International guidelines recommend continuing anticoagulant treatment for patients with an unprovoked VTE or provoked VTE with a persisting risk factor, unless individual bleeding risks are deemed high, and advocate limiting treatment duration in case of transient provoking risk factors [13]. Our results support that, for the majority of PWH suffering a VTE, current recommendations can be considered appropriate for PWH with a provoked or unprovoked first VTE as well. Indeed, in a majority of the recurrences suffered, there was no clear identifiable HIV-related risk factor.

However, our data suggest that for a subgroup of PWH who were immunocompromised at the first VTE, the risk of VTE recurrence is lower when the immune status recovers during anticoagulant treatment. Considering immunodeficiency during a first VTE as a reversible risk factor specific to PWH fits with the current theoretical framework regarding influence of reversible risk factors on VTE recurrence risk [14]. Our findings support exploring an opportunity to limit the duration of anticoagulant therapy in PWH who have a sufficiently reversed immunodeficient state that was initially present at time of the first, otherwise unprovoked, VTE.

To our knowledge, our study is the only study currently available systematically assessing VTE recurrence risk in PWH and providing guidance for their anticoagulation management after first VTE. A main strength is the use of a large representative sample of PWH and a large cohort of controls from a comparable geographical region. The large number of participants also enabled us to adequately adjust for the most important confounders. Furthermore, we consider the possibility for detailed validation of events in our study a particular strength, as such validation may not always be possible when using large-scale cohort studies that were not dedicated to assessing VTE as an outcome [10].

Several limitations should be considered. Given that we found a higher overall risk of recurrent VTE for PWH compared to controls, our findings would be problematic if there was some systematic bias overestimating recurrent VTE in PWH compared to controls. With

regard to ascertainment of the outcome, there are 3 possible sources of bias: adjudication methods, loss to follow-up, and period effects.

In the PWH cohort, the main concern in the context of our findings is whether there is inherent overestimation of recurrence risk. With respect to adjudication, it suffices to say that all certain events were adjudicated on the basis of clinical records; imaging reports were unequivocal about location of events. In fact, the 4 possible recurrences were adjudicated as such because the clinical records were insufficient to conclude a certain recurrence; this was mainly due to the notion that these patients had their diagnostic procedures done elsewhere than in their respective HIV centres. In these possible cases of recurrence, clinical records received from elsewhere were too scant to adjudicate a certain recurrence. Therefore, this issue is likely to have underestimated recurrence risk in the PWH cohort.

Loss to follow-up in the PWH cohort occurred in only 5 participants. Two participants emigrated and three others were not retained in care. Whilst emigrating is likely noninformative, not being retained in care implicitly means suboptimal treatment of HIV. So, this source of loss to follow-up is likely to be associated with a higher risk of recurrence. Here again, this source of bias likely means we are slightly underestimating recurrence risk in the PWH, again reassuring in the light of our findings. Finally, we addressed period effects in sensitivity analyses—as stated, there is potential for overestimation of VTE recurrence in the PWH cohort due to diagnosis of subsegmental PE. However, we only found one recurrent event, which was an isolated subsegmental PE. This did not materially change results from the main analysis.

These sources of bias given earlier should also be considered in the MEGA cohort. Given our results, it would be concerning if there were bias leading to underestimation of VTE recurrence. With regard to adjudication, it is plausible that underestimation might have occurred, as this was dependent on centres in which patients had received their diagnosis responding to information and subsequently providing sufficient data on VTE diagnosis (see S1 Supplementary Methods for adjudication criteria). However, when this was not possible, the adjudication criteria permitted adjudication of a certain event if patient and their anticoagulation clinic reported an event that was clearly in a different location from their first VTE. The sensitivity analysis combining certain and possible recurrences is illustrative in this respect: possible recurrences in MEGA could here be adjudicated liberally—based solely on patient self-report. It is important to note that this was not possible in ATHENA, therefore this sensitivity analysis likely represents a worst-case scenario with regard to balance in adjudication in favour of the MEGA cohort. Even under these circumstances, the HR estimates remain above 1, with lower bound of the confidence intervals excluding an importantly lower risk of recurrence for PWH.

With regard to loss to follow-up in MEGA, this was mainly driven by nonresponse, which occurred in about 10% of participants. Contrary to ATHENA, there is no clear indication that this may have been driven by clinical risk. Examination of characteristics of nonresponders reveals that the proportion of participants with a provoked event was higher compared to the overall cohort (see S1 Supplementary Analyses). This suggests that lower-risk patients were less likely to respond, which is plausible: participants without a recurrence may have had a slight lower tendency to respond to the investigators. This would suggest that there is potential for overestimation of recurrence risk, which is again reassuring in the light of our findings.

A final potential concern would be that the different definitions of exposure between cohorts may have led to mis-specification of the effect of confounders, leading to insufficient adjustment of the HR relating to HIV infection. We addressed this issue in our sensitivity analyses, taking worst-case scenarios into account; the effect estimates either made the association stronger (defining hospitalisation as provoked/unprovoked) or only changed the association downwards minimally (cancer sensitivity analysis).

With regard to the analysis of HIV-specific factors, the finding of a lower risk of recurrence associated with better immune recovery would be problematic if there were systematic bias toward underestimating VTE in these PWH specifically. However, a pilot study prior to major study roll-out had 100% sensitivity showing that missing VTE by our strategy in ATHENA is highly unlikely. Furthermore, immunodeficient PWH are generally monitored more frequently, which makes missing a recurrent VTE risk in this population even more unlikely; indeed, the opposite is more likely to be the case. Finally, the number of PWH at risk became smaller with longer follow-up, consequently leading to more uncertainty of the observed estimates, precluding firmer conclusions on the risk of recurrent VTE in relation to CD4+ T-cell recovery in PWH with otherwise unprovoked first VTE, as treatment advice in guidelines is driven by absolute and not relative risks [13]. Therefore, with the current data, we cannot make firm, practical recommendations on the decision to continue or withhold treatment in this situation, but these data should be seen as supportive in making a decision.

To further elucidate the influence of HIV on VTE incidence, we hope that other HIV cohorts will replicate our evaluation and connect for meta-analysis. A relevant research aim arising from our study is a, preferably prospective, determination of what CD4+ T-cell recovery threshold is sufficient to consider discontinuing anticoagulants in PWH with otherwise unprovoked first VTE. Follow-up studies should also preferentially include PWH with different characteristics, including being female, having a varying transmission mode (IV drug use), or being from sub-Saharan Africa regions; factors such as younger age, more balanced sex ratio, and higher burden of concomitant infections (e.g., malaria, tuberculosis) among affected populations may further influence VTE risk. Finally, to fully inform net benefits of continuing or withholding anticoagulation in PWH with VTE, studies should also preferably evaluate the absolute risk of major bleeding associated with prolonged anticoagulant use among PWH.

In conclusion, we observed a high risk of VTE recurrence in PWH. Clinicians managing VTE in PWH should incorporate all relevant data to decide on anticoagulant treatment duration and do so in light of the full clinical profile of individual patients. As an overall message, in line with international guidelines, anticoagulant therapy should be continued in most PWH after an unprovoked first VTE and stopped after a provoked first VTE [13]. However, immunocompromised PWH with an otherwise unprovoked first VTE and substantial CD4+ T-cell recovery during anticoagulant treatment appear to have a reversible provocative risk factor. This is of relevance when deciding on (dis)continuing anticoagulant therapy in PWH with otherwise unprovoked first VTE.

## Supporting information

**S1 STROBE Checklist.**
(DOC)

**S1 Protocol.**
(DOCX)

**S1 Supplementary Methods.**
(DOCX)

**S1 Supplementary Analyses.**
(DOCX)

**S1 Fig. Cumulative incidence of recurrent VTE stratified by HIV status and sex.**
(TIF)

**S2 Fig. Cumulative incidence of recurrent VTE stratified by HIV status and unprovoked versus provoked index VTE.**
(TIF)

## Acknowledgments

We would like to acknowledge the data monitoring staff as well as central coordinating staff at the HIV Monitoring Foundation (which manages ATHENA), who supported data collection on VTEs, and thank them for responding to extra data queries in a timely fashion. We would like to thank S. Kochen, medical student, for collecting data. We also thank Dr. M. E. van der Ende for her help in conceptualising the study idea.

## Author Contributions

**Conceptualization:** Casper Rokx, Jaime F. Borjas Howard, Peter Reiss, Suzanne C. Cannegieter, Karina Meijer, Wouter Bierman, Vladimir Tichelaar, Bart J. A. Rijnders.

**Data curation:** Jaime F. Borjas Howard, Colette Smit, Elise D. Pieterman, Willem M. Lijfering.

**Formal analysis:** Jaime F. Borjas Howard, Ferdinand W. Wit, Elise D. Pieterman, Willem M. Lijfering.

**Funding acquisition:** Peter Reiss, Suzanne C. Cannegieter.

**Investigation:** Casper Rokx, Jaime F. Borjas Howard, Peter Reiss, Suzanne C. Cannegieter, Wouter Bierman, Vladimir Tichelaar.

**Methodology:** Casper Rokx, Jaime F. Borjas Howard, Colette Smit, Ferdinand W. Wit, Suzanne C. Cannegieter, Willem M. Lijfering, Karina Meijer, Bart J. A. Rijnders.

**Project administration:** Casper Rokx, Jaime F. Borjas Howard, Elise D. Pieterman.

**Supervision:** Colette Smit, Ferdinand W. Wit, Willem M. Lijfering, Karina Meijer, Vladimir Tichelaar, Bart J. A. Rijnders.

**Writing – original draft:** Casper Rokx, Jaime F. Borjas Howard.

**Writing – review & editing:** Casper Rokx, Jaime F. Borjas Howard, Colette Smit, Ferdinand W. Wit, Elise D. Pieterman, Peter Reiss, Suzanne C. Cannegieter, Willem M. Lijfering, Karina Meijer, Wouter Bierman, Vladimir Tichelaar, Bart J. A. Rijnders.

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
