## [Decision Letter · Decision Letter 0]

20 Jan 2020

Dear Dr. Borjas Howard,

Thank you very much for submitting your manuscript "The risk of recurrent venous thromboembolism in patients with HIV: a nationwide cohort study." (PMEDICINE-D-19-04164) for consideration at PLOS Medicine. 

Your paper was discussed among the editorial team and sent to independent reviewers, including a statistical reviewer. The reviews are appended at the bottom of this email and any accompanying reviewer attachments can be seen via the link below:

[LINK]

In light of these reviews, we will not be able to accept the manuscript for publication in the journal in its current form, but we would like to invite you to submit a revised version that fully addresses the reviewers' and editors' comments. You will appreciate that we cannot make a decision about publication until we have seen the revised manuscript and your response, and we expect to seek re-review by one or more of the reviewers. 

We hope to receive your revised manuscript by Feb 10 2020 11:59PM. Please email us (plosmedicine@plos.org) if you have any questions or concerns.

Please let me know if you have any questions. Otherwise, we look forward to receiving your revised manuscript in due course. 

Sincerely,

Richard Turner, PhD

rturner@plos.org

You will note that one referee questions the validity of the study, and we suggest a detailed response on this point. 

In your abstract and elsewhere, please add p values alongside 95% CI where available. 

Please add summary demographic details for study participants to the abstract. 

Please add a new final sentence to the "methods and findings" subsection of your abstract, which should summarize the study's main limitations. 

After the abstract, we will need to ask you to add a new and accessible "author summary" section in non-identical prose. You may find it helpful to refer to one or two recent research papers published in PLOS Medicine to get a sense of the preferred style. 

Early in the methods section of your main text, please state whether or not the study had a protocol or prespecified analysis plan, and if so attach the relevant document(s) as a supplementary file (referred to in the text). Please highlight non-prespecified analyses. 

Please briefly note that informed consent was obtained, around line 122. 

Please remove the "role of the funding source" statement from the text (the relevant information will appear in metadata in the event of publication). 

At line 377, please adapt the text to begin "In this study, we found that ... were at higher risk ..." or similar. 

At line 396, please add "to our knowledge" or similar. 

Throughout your text, please do not use italics for emphasis.

At line 452 and any other relevant instances, please adapt the wording to avoid overstating the conclusions (e.g., "In conclusion, we observed a high risk of VTE recurrence in PWH.").

Throughout the paper, please format reference call-outs as follows: "... VTE recurrence [19,20].".

We ask you to revise your reference list to ensure that citations meet journal format. For example, italics should be removed; and 6 rather than 3 author names should be listed where appropriate prior to "et al.".

Please adapt the attached STROBE checklist so that individual items are referred to by section (e.g., "Methods") and paragraph number rather than by page or line numbers, as the latter generally change in the event of publication. Please refer to the document in the methods section. 

Comments from the reviewers:

*** Reviewer #1: 

I confine my remarks to statistical aspects of this paper. These were very well done and I recommend publiucation

I had one question - why did the authors choose to control for variables rather than do matching or propensity scores?

Also, i commend the authors for keeping variables contininous but presenting them stratified.

Peter Flom

*** Reviewer #2: 

The risk of recurrent VTE in PWH is a significant question in clinical practice, especially in an unprovoked first event. This because deciding whether patients should receive indefinitive anticoagulation or stop treatment as current guidelines advocate is a crucial step.

We found this this manuscript well written and organized, and capable somehow to respond to the question about risk rate of VTE recurrence in PWH further suggesting that a rapid and well reconstituded CD4+T-cell count may favor discontinuation of anticoagulation.

However we found several unresponded questions and evidence

1) Has the role of DOACs and their interactions with modern ART therapy been assessed? 

2) has the aderence to ART therapy during anticoagulation been assessed?

3) Patients without suppressed HIV-RNA should not be included in the study 

4) We do not agree to rerun models considering only partecipants with CD4+T-cell below 500 at the time of index VTE, what the real impact of the dynamics of CD4+ recovery on the risk? 

*** Reviewer #3: 

Rokx et al have conducted an important study comparing the outcome of HIV patients with VTE to patients without HIV. As suggested by the authors these result have significant implications for treatment of patients with HIV and VTE. They compare outcomes in two different cohorts; the ATHENA cohort and the MEGA study cohort. Although both these studies were conducted in the Netherlands, data was collected on participants at different times (ATHENA 2003-2018; MEGA 1999-2009) therefore it is possible, as noted by the investigators, that improved diagnostic methods (e.g., multidetector CT angiography, more sensitive D dimer assays) may contributed to the differences in outcomes noted. In this light, do the authors have additional information on the number of subsegmental PE in the two groups at initial diagnosis and follow up? More detailed information on the types of VTE (upper versus lower extremity DVT, lobar versus segmental vs subsegemental PE) might provide additional evidence to support their conclusion that HIV patients have an increased risk for recurrent VTE. 

The authors finding that greater CD4 recovery was associated with a lower incidence of recurrent VTE is thought provoking. Do the authors have any information on when patients started ART? One wonders if initiation of ART at the time of VTE diagnosis is associated with a reduced risk for recurrent VTE. 

In the discussion I think it would be important to point out that only a small number of the PWH acquired their infections via IV drug use. I suspect that these patients may have substantially different rate of recurrent VTE than PWH acquired via sexual transmission. Studies in PWH acquired via IV drug use will be important to conduct to determine if their outcomes are similar. 

Minor issues

On page 16, line 321, there is a typo. I would remove "5-"

On page 18, line 353, the sentence might be clearer if written as follows "Within PWH who suffered [recurrent VTE}, over 65% had a CD4+ T-cell count over 500 and 83% had an undetectable viral load at the time of their recurrence."

*** Reviewer #4: 

The authors aimed to estimate the risk of recurrent VTE in PWH compared to controls, and to identify risk factors for VTE recurrence within this population.

The most severe problem is in the study design, which could invalidate the study. The authors used two cohorts with different characteristics:

- These two cohorts are from two different periods (MEGA: 1999-2009, ATHENA: 2003-418 2018), in which the diagnostic practices could have an influence. 

- The sample size in the two cohorts is very different. Particularly problematic is the small sample size of ATHENA cohort.

- These two cohorts have different definitions of exposures.

- The outcome variable (recurrent VTE) is collected differently in the two cohorts.

Minor comments:

The reference list should be expanded and updated (more recent publications should be included).

There are some mistakes:

- Line 75: "had lower plasma CD4+T-cell counts".

- Line 105: "supported by recent a recent study showing". 

- Line 302: HIV specific characteristics. Align the data with the PWH column.

- Line 346: "which persisted over time (fig.2)". Figure 2 is not included in the article.

Review all the text to eliminate this type of errors.

Major comments:

a) The percentage of women is double in the MEGA cohort. It is not a good control group. The authors should select a subpopulation that has a similar percentage of women.

b) Are lost and excluded patients similar to those included in the study?

c) CD4+ T-cells at baseline were related to lower risk of VTE recurrence. How do they explain this finding? It does not seem to make sense, since low CD4+ values have been associated with a higher risk of the first VTE. In addition, immunopathological data support this association, since at lower CD4+ values, more significant inflammation and more risk of VTE.

***

[LINK]

---

## [Decision Letter · Decision Letter 1]

26 Feb 2020

Dear Dr. Borjas Howard,

Thank you very much for re-submitting your manuscript "The risk of recurrent venous thromboembolism in patients with HIV: a nationwide cohort study." (PMEDICINE-D-19-04164R1) for consideration at PLOS Medicine.

I have discussed the paper with editorial colleagues, and it was also seen again by three reviewers. I am pleased to tell you that, provided the remaining editorial and production issues are dealt with, we expect to be able to accept the paper for publication in the journal.

[LINK]

Please let me know if you have any questions. Otherwise, we look forward to receiving the revised manuscript shortly. 

Sincerely,

Richard Turner, PhD

rturner@plos.org

Requests from Editors:

Please revisit your data statement. PLOS policy is that data should be made available without restriction, except for considerations regarding ethics approval, say. We suggest truncating the current statement at " ... lumc.nl)." and are happy to discuss this further as needed. 

We recognize the debate regarding explicit provision of p values, and again ask that these are included in abstract and main text, where available. 

Please make that "Risk of recurrent venous thromboembolism in people with HIV infection ..." in the title. 

At line 51, please make that "median age 48 years".

At line 62, please amend the text to "The main study limitations are that ..." or similar. We ask you to move the subsequent sentence regarding sensitivity analyses earlier in the "methods and findings" subsection, as the sentence on limitations should conclude this subsection. 

At line 65, please make that "principal analyses".

At line 70, we suggest avoiding the wording "the data strongly suggested" in favour of "... among PWH, recurrence risk appeared to decrease with greater ..." or similar. 

At line 71, we suggest amending "(dis)continue" to "... when deciding whether or not to discontinue ..." or similar. Please also amend the text at line 554.

At line 86, please remove the word "already". 

At line 101, please make that "fewer recurrent events". 

At line 106, we ask you to amend the text to "... is apparently increased in PWH", or similar. 

Around line 133, please add one or more references to the "current guidelines". Please also consider this at line 550.

At line 167, would that be written informed consent?

At line 332, are you able to amend the text to "... All analyses were prespecified except several analyses that were suggested by ..."?

Please adapt the text around line 336 to "... (see S1_Checklist)" or similar. 

At line 548, we suggest substituting "elevated" for "high".

Throughout the text, please relocate reference call-outs according to the following style: "... described previously [17-19].".

In the reference list, please ensure that all journal names are abbreviated as appropriate, e.g., "PLoS Med." for reference 17.

Comments from Reviewers:

*** Reviewer #1: 

I had no problems with the earlier draft and I recommend publication

Peter Flom

*** Reviewer #2: 

After reading the authors' comments and additions to my questions and requests from other reviewers, I must congratulate everyone on the work they have done to make this publication of substantial value.

***

[LINK]

---

## [Editor Report · Decision Letter 2]

13 Apr 2020

Dear Dr Borjas Howard, 

On behalf of my colleagues and the academic editor, Dr. Salvador Resino, I am delighted to inform you that your manuscript entitled "Risk of recurrent venous thromboembolism in patients with HIV: a nationwide cohort study." (PMEDICINE-D-19-04164R2) has been accepted for publication in PLOS Medicine. 

PRODUCTION PROCESS

PRESS

PROFILE INFORMATION

Thank you again for submitting the manuscript to PLOS Medicine. We look forward to publishing it. 

Best wishes, 

Richard Turner, PhD

Senior Editor 

PLOS Medicine

plosmedicine.org